# New Insight into Assembled Fe_3_O_4_@PEI@Ag Structure as Acceptable Agent with Enzymatic and Photothermal Properties

**DOI:** 10.3390/ijms231810743

**Published:** 2022-09-15

**Authors:** Teng Wang, Xi Hu, Yujun Yang, Qing Wu, Chengdian He, Xiong He, Zhenyu Wang, Xiang Mao

**Affiliations:** 1State Key Laboratory of Ultrasound in Medicine and Engineering, College of Biomedical Engineering, Chongqing Medical University, Chongqing 400016, China; 2Chongqing Key Laboratory of Biomedical Engineering, Chongqing Medical University, Chongqing 400016, China; 3Key Laboratory of Laboratory Medical Diagnostics, Ministry of Education, Department of Laboratory Medicine, Chongqing Medical Laboratory Microfluidics and SPRi Engineering Research Center, Chongqing Medical University, Chongqing 400016, China

**Keywords:** Fe_3_O_4_@PEI@Ag, assembling process, enzymatic, catalytic performance, photothermal property

## Abstract

Metal-based enzyme mimics are considered to be acceptable agents in terms of their biomedical and biological properties; among them, iron oxides (Fe_3_O_4_) are treated as basement in fabricating heterogeneous composites through variable valency integrations. In this work, we have established a facile approach for constructing Fe_3_O_4_@Ag composite through assembling Fe_3_O_4_ and Ag together via polyethyleneimine ethylenediamine (PEI) linkages. The obtained Fe_3_O_4_@PEI@Ag structure conveys several hundred nanometers (~150 nm). The absorption peak at 652 nm is utilized for confirming the peroxidase-like activity of Fe_3_O_4_@PEI@Ag structure by catalyzing 3,3′,5,5′-tetramethylbenzidine (TMB) in the presence of H_2_O_2_. The Michaelis–Menten parameters (K_m_) of 1.192 mM and 0.302 mM show the higher catalytic activity and strong affinity toward H_2_O_2_ and TMB, respectively. The maximum velocity (V_max_) value of 1.299 × 10^−7^ M∙s^−1^ and 1.163 × 10^−7^ M∙s^−1^ confirm the efficiency of Fe_3_O_4_@PEI@Ag structure. The biocompatibility illustrates almost 100% cell viability. Being treated as one simple colorimetric sensor, it shows relative selectivity and sensitivity toward the detection of glucose based on glucose oxidase. By using indocyanine green (ICG) molecule as an additional factor, a remarkable temperature elevation is observed in Fe_3_O_4_@PEI@Ag@ICG with increments of 21.6 °C, and the absorption peak is nearby 870 nm. This implies that the multifunctional Fe_3_O_4_@PEI@Ag structure could be an alternative substrate for formatting acceptable agents in biomedicine and biotechnology with enzymatic and photothermal properties.

## 1. Introduction

Transition metal-based functional materials exhibit unique properties that could be ascribed to their structural electronic status or elemental integrations [1,2,3,4]. Particularly, they demonstrate high quality and use in various applications according to different requirements in oxide or dioxide compositions [5,6]. Conventionally, they convey particular characterization because of the d-electrons effect in full utilizations. It is worth noting that iron oxide nanoparticles (Fe_3_O_4_ NPs) are fundamental transition oxides, with expected applications as magnetic, biocompatible, and nontoxic agents [7,8]. In applied strategies, Fe_3_O_4_ NPs possess intrinsic reaction activities because of their metal bond capacity and variable valency [9]. In previous works, the characteristics of Fe_3_O_4_ NPs have mainly been investigated in enzymatic performances at initial times [10]. These pioneering works improved the possibility that Fe_3_O_4_ materials could integrate other functional agents as novel types of enzyme mimics. In these comparisons, Fe_3_O_4_ series enzyme mimics showed high stability and efficiency in catalytic performances, the same as natural enzymes exhibit under harsh reaction conditions [9,10,11]. Moreover, they reflected that Fe_3_O_4_ NPs have potential capacity and were considered as a competitive agent in forming heterogeneous fabrications. Additionally, the regular requirements for the synthetic procedure, structure design, composition flexibility, and customizable catalytic activity are definitely determined by adjusting the reaction atmospheres based on preparing approaches [12,13,14]. Furthermore, the related applications of Fe_3_O_4_ NPs include wide use in cancer therapy [15,16,17,18], immunotherapy [19,20,21], biosensors [22,23,24], catalysis [25,26,27], and drug delivery [28,29,30]. Therefore, it is essential to explore multifunctional or heterogeneous composites that are treated as metal enzyme mimics and maintain their catalytic activities completely.

In order to avoid the limitations of enzymatic activity, Fe_3_O_4_ NPs were utilized in constructing heterogeneous enzymatic agents toward various fields [13,21,25,31,32,33,34]. According to the function requirements, noble metal elements were always selected as combiners for making heterogeneous structures. They have illustrated their constructed combing complexes and additional physical or chemical properties, besides obtaining the original characteristics of magnetic composites. Regarding the individual noble metal, it gives excellent catalytic activity and optical properties because of its flexible electron transmission between metal atoms and outstanding stabilization [35,36,37,38,39]. Different metal forms have been used, and heterogeneous structures have been synthesized by combining them with Fe_3_O_4_ NPs. These heterogeneous enzyme mimics include Pd@Fe_3_O_4_-MWCNT nanocomposite [31], Au@Fe_3_O_4_ MOF [32], Fe_3_O_4_-Ag [33], and Pt-Fe_3_O_4_ [34]. These studies have discovered that doping noble metals into Fe_3_O_4_ NPs can improve or expand their enzymatic activities. The fundamental reason for this could be ascribed to both doped noble metals and Fe_3_O_4_ NPs, which have a synergetic catalysis effect and lead to the enhancement of catalytic performances. Fe_3_O_4_-Ag fabrication was used in antibacterial works, and the enzymatic process also showed a synergetic catalysis effect which enhanced the antibacterial effect due to the materials’ activity [19,33,40,41,42,43]. It required the high uniformity and adjusted size distribution of heterogeneous agents. The synthesis of the small size and uniformity of Fe_3_O_4_-Ag require a relatively strict preparation process and harsh preparation conditions. Wei et al. [32] prepared Fe_3_O_4_@MoS_2_-Ag with a defect-rich rough surface by a Teflon-autoclave at 180 °C for 10 h and in-situ photo deposition of Ag NPs. It exhibited acceptable antibacterial performance and excellent peroxidase-like activity. Pan et al. [19] synthesized Ag_x_Au_y_/Fe_3_O_4_ composition via Teflon-lined stain steel autoclave, and sealed and maintained it at 150 °C for 20 h. The results showed that the activity of Ag_x_Au_y_/Fe_3_O_4_ NPs were improved along with increasing Ag content. The hydrophobic individual content (Fe_3_O_4_ and Ag NPs) exhibits negative influences while utilizing them in further biological applications [40,44]. It needed further modification in order to apply its original physical and chemical properties. In the modification procedures, the relevant properties of the enzyme were inevitably lost, and the preparation process was more complicated. This conveyed the main clues of synthetic approaches, which indicated that hydrophilic synthesis would be a preferred option in fabricating heterogeneous constructions. Therefore, the preparation of hydrophilic Fe_3_O_4_ and Ag composites as enzyme mimics by a simple chemical method is still a great challenge.

Herein, we reported one facile approach to make Fe_3_O_4_@PEI@Ag structures as acceptable agents with enzymatic and photothermal properties. Heterogeneous Fe_3_O_4_@PEI@Ag structures were fabricated by using hydrophilic Fe_3_O_4_ NPs as hard templates and were surface-modified by PEI molecules. Ag atoms gradually anchor on the Fe_3_O_4_ surface and the assembled Fe_3_O_4_@PEI@Ag structure is constructed in water medium. Enzymatic performance was characterized as peroxidase-like activity, which verifies and illustrates in colorimetric reaction with TMB molecule. It also revealed that the heterogeneous Fe_3_O_4_@PEI@Ag structure has a sensitive response to glucose colorimetric reactions, and can be applied to the detection of human diabetes. By subsequently coating them with ICG molecule, the Fe_3_O_4_@PEI@Ag@ICG structure not only saw severely enhanced photothermal properties (at 808 nm irradiation) and fluorescence imaging applications [44], but also the greatly improved stability of both parties, thanks to this combined structure. Furthermore, this heterogeneous structure could have multidirectional applications in different applications eventually.

## 2. Results and Discussion

### 2.1. The Synthesis Work of Fe_3_O_4_@PEI@Ag, Fe_3_O_4_@PEI@Ag@ICG, and Relative Structural Characterizations

There are two kinds of Fe_3_O_4_-based heterogeneous structures that were fabricated: Fe_3_O_4_@PEI@Ag and Fe_3_O_4_@PEI@Ag@ICG structures. There have unique influences on synthesis processes while using PEI and ICG molecules, respectively. In this work, the assembled structures were synthesized using a “layer-by-layer” method, as shown in Figure 1. The characteristics of the synthesized Fe_3_O_4_@PEI@Ag structure are depicted in Figure 1a. Firstly, the wet chemistry method was used to synthesize Fe_3_O_4_ and the amino-functionalized Fe_3_O_4_ NPs (10 nm, Appendix A) could be obtained by using PEI for additional modifications. Further, the hydrophilic PEI with the amino group could assemble on the surface of Fe_3_O_4_ under mechanical stirring in order to obtain a highly dispersed Fe_3_O_4_@PEI structure. By mediating Ag growth and anchoring it to make heterogeneous structures, PEI-mediated seed growth method was used to form Fe_3_O_4_@PEI@Ag and Fe_3_O_4_@PEI@Ag@ICG structures, respectively. Fe_3_O_4_@PEI seeds were used as nucleation sites to coat the Ag by reducing silver precursors (C_2_H_3_O_3_Ag). Among the deposition procedures of metal precursors, Ag seeds could immobilize on the surface of Fe_3_O_4_@PEI through electrostatic interactions. The individual Ag seeds were approximately 10 nm (Appendix A) and were tightly attached to the Fe_3_O_4_@PEI surface. The main charged ability of NPs’ surface can be attributed to the PEI’s structural characteristics. Amino groups promoted a strong positive charge and increased the combination for assembling Fe_3_O_4_ and Ag completely. The diameter of the resulting Fe_3_O_4_@PEI@Ag structure is about 150 nm and exhibited good dispersion in aqueous solution. Subsequently, the negatively charged ICG molecules were uniformly attached to the surface of Fe_3_O_4_@PEI@Ag structure in the gap of the Ag particles. It could be thought of as one assembling procedure between two different particles (Figure 1a,b). Elemental mapping analysis was performed to confirm the elemental composition and illustrate the distribution of the Fe_3_O_4_@PEI@Ag structure, as shown in Figure 1b–f. This proved that the assembly process can be well-realized in the form of regular aggregation.

As shown in Figure 1g, the UV-vis spectrum reflected the different optical absorbance of pure phased Fe_3_O_4_, Ag, and Fe_3_O_4_@PEI@Ag structures, respectively. Here, it is observed that this difference was obvious at the absorption spectrum, conveying that Ag showed surface plasmon resonance (SPR) at nearby 415 nm, which indicated its characteristic of a smaller diameter, less than 10 nm [43]; however, Fe_3_O_4_ NPs did not have a SPR effect, exhibiting a downward curve. There was an additional phenomenon in the resulting absorption of the Fe_3_O_4_@PEI@Ag structure, which exhibited a higher SPR band at 600 nm, confirming that the Ag NPs were successfully attached to Fe_3_O_4_ with red-shifting phenomenon. This can be attributed to the plasmon resonance effects of the formed flower-structured Ag shell [19]. Furthermore, the SPR phenomenon appeared gradually while the Fe_3_O_4_@PEI@Ag structure formed the accumulation of the SPR effect in a single metal and can induce new aggregation effect on structures [45]. We tried to use PVP and PEG instead of the PEI molecule in characterization, as shown in Appendix A. When PVP was used, it could link Ag and Fe_3_O_4_ together, but the UV characteristic peak at 415 nm disappeared after being left for some time, which proved that using PVP as the linker was not stable. When PEG was used, there was no characteristic also in optical measurements, which might prove that PEG was not suitable as a linker for Fe_3_O_4_ and Ag additionally. XRD characterization was applied to analyze the crystalline structure of Fe_3_O_4_ and Fe_3_O_4_@PEI@Ag NPs, as shown in Figure 1h. The XRD pattern of the Fe_3_O_4_ NPs showed six characteristic diffraction peaks at 30.1°, 35.5°, 43.3°, 53.7°, 57.2°, and 62.8° corresponding to the (220), (311), (400), (422), (511), and (440) crystal planes of pure magnetite (Fe_3_O_4_) with a cubic spinel structure (JCPDS card. No. 03-065-3107). The high crystallinity of the Fe_3_O_4_ was evident from the sharp diffraction pattern peaks. On the other hand, the XRD pattern of Fe_3_O_4_@PEI@Ag was similar to that of Fe_3_O_4_ particle, and the pattern of the Fe_3_O_4_@PEI@Ag shows additional peaks at 2θ values of 38.1°, 44.3°, 64.5°, 77.4°, corresponding to the reflections of the (111), (200), (220), (311) crystalline planes, respectively, of the face-central cubic crystal structure of Ag. The above result was consistent with ASTM standard (JCPDS Card No. 04-0783) and further confirmed that the Ag on the surface of the composite structure exists in the zero-valent state. FT-IR is an appropriate technique to study chemical adsorption or chemical interaction [44]. As shown in Figure 1i, FT-IR spectrum conveyed the pure Fe_3_O_4_, Fe_3_O_4_@PEI, and Fe_3_O_4_@PEI@Ag structure, respectively. According to Fe_3_O_4_ particles, the peak around 577 cm^−1^ belonged to Fe-O vibration [45]. The peak absorption bands at 3405 cm^−1^ and 1627 cm^−1^ correspond to the O-H stretching vibration and bending vibration, respectively [46,47,48]. After PEI is coated on the surface of Fe_3_O_4_, the peak at 1546 cm^−1^ in the Fe_3_O_4_@PEI curve represents the N-H asymmetric bending vibration peak, and the Fe-O stretch at 577 cm^−1^ was obviously weakened. The appearance and vibrations of the functional groups indicated that the Fe_3_O_4_ NPs were modified by PEI through a covalent bond [49]. Meanwhile, the incorporation of Ag did not change the structures of the composites. All of the above implied the linkage between particles and molecules were exactly impressed.

We aimed toward a Fe_3_O_4_@PEI@Ag structure, which could be considered as a magnetic carrier. Conventionally, a magnetic carrier is useful for directing the spatial distribution of drugs via noninvasive magnetic guiding. To investigate the magnetic response of the Fe_3_O_4_ and Fe_3_O_4_@PEI@Ag structure, the magnetic properties of different samples were investigated at 300 K and the superparamagnetism of Fe_3_O_4_@PEI@Ag had a lower saturation magnetization of 8.50 emu∙g^−1^ compared to the pure Fe_3_O_4_ (57.04 emu∙g^−1^). The reduced magnetism of Fe_3_O_4_@PEI@Ag structure was due to the compact shielding of nonmagnetic PEI and Ag on the Fe_3_O_4_ surface, which could be explained by the lower mass ratio of Fe_3_O_4_ in-unit composites [50]. Additionally, we used a magnet placed next to the sidewall of a cuvette containing an aqueous dispersion (10 mg∙mL^−1^) of Fe_3_O_4_@PEI@Ag. It can rapidly migrate toward the magnet and is accumulated against the inner wall within 1 min (the Fe_3_O_4_ NPs just needed 30 s) as illustrated in Appendix A, and they could be dispersed again homogeneously if the magnet was removed (Figure 2a). Figure 2b showed the changes in the ζ-potential measurement of each component while mediating with Fe_3_O_4_, which can monitor the modification in fabricating assembled Fe_3_O_4_@PEI@Ag structures. In detail, the changes in surface potential from negative to positive after PEI coated, which demonstrated the successful assembly of cationic PEI on Fe_3_O_4_ NPs by electrostatic interaction. At same time, the zeta potential value changed from−17.99 mV to +20.70 mV. Once Ag was adhered to formation, it could increase to +25.11 mV, which proved the dispersion (Fe_3_O_4_@PEI@Ag structure) was improved compared to Fe_3_O_4_@PEI completely. Similarly, the zeta potential turned negative again (+25.11 mV to −14.95 mV) due to the successful conjugation of negatively charged ICG molecules on the surface of the Fe_3_O_4_@PEI@Ag structure. In the PBS environment, it can improve the dispersion of the Fe_3_O_4_@PEI@Ag@ICG structure (−14.95 mV to −22.21 mV). Before and after being modified by the ICG molecule, the stability of the Fe_3_O_4_@PEI@Ag structure was also tested as shown in Appendix A; it maintained good dispersion in solvent even after 48 h of culturing.

### 2.2. Peroxidase-like Activity of Assembled Fe_3_O_4_@PEI@Ag Structure

Originally, Fe_3_O_4_ NPs possess intrinsic peroxidase-like activities under acidic conditions in previous studies [7]. The typical reaction of TMB and H_2_O_2_ were used to investigate the peroxidase-like activity of Fe_3_O_4_@PEI@Ag structure. As shown in Figure 3a, pure phased Fe_3_O_4_ (100 μL, 100 μg∙mL^−1^), Ag (100 μL, 100 μg∙mL^−1^), and Fe_3_O_4_@PEI@Ag structure (100 μL, 100 μg∙mL^−1^) were measured to check their characteristics after reaction with H_2_O_2_ (100 μL, 1 mM) and TMB (100 μL, 1 mM) within 10 min (UV-vis spectra). The inset images showed typical digital images of each colloidal solution. The peaks matched, being nearly about 652 nm, which implied the intrinsic peroxidase-mimicking activity of each colloidal solution [51,52]. It implied that the catalytic reaction to oxidize TMB corresponded to TMBox by H_2_O_2_. Comprehensively, the typical absorbance at 652 nm was observed with 5 min incubations, which is completely different from control works. In these, the target TMB could not show any changes in colors. It is worth noting that the Fe_3_O_4_@PEI@Ag structure enhanced peroxidase-like activity, even as each (Fe_3_O_4_ and Ag) exhibited relative catalytic properties. Here, the higher catalytic performance could be ascribed to synergistic effects from the assembling process of Fe_3_O_4_ and Ag NPs. This enzymatic property was investigated under different pH conditions. As shown in Appendix A, pure Fe_3_O_4_ NPs were firstly studied in the catalytic reaction of the TMB vs. pH value, and they exhibited catalytic activity only at pH = 5.8. Likewise, the Fe_3_O_4_@PEI@Ag structure exhibited the highest catalytic performance at pH = 5.8 as shown in Figure 3b. With the increasing pH or the decreasing pH, the relative activity of the Fe_3_O_4_@PEI@Ag structure decreased even without enzyme activity. The peroxidase-like activities of Fe_3_O_4_@PEI@Ag structure could gradually be enhanced by increasing the amount of the catalyst, as shown in Figure 3c. We found that the reaction rate of TMB oxidation was highly dependent on the amount of Fe_3_O_4_@PEI@Ag structure. In Figure 3d, the strong absorbance at 652 nm could well prove the advanced catalytic reaction within 30 min at RT, and it exhibited the catalytic performance of the assembled Fe_3_O_4_@PEI@Ag structure. The intensity of this was recorded and is shown in Appendix A. As the reaction time increased, the oxidation products in the products could gradually accumulate together more than before.

### 2.3. Steady-State Kinetic Assay of Fe_3_O_4_@PEI@Ag Structure

The Michaelis–Menten equation was used to investigate the kinetics of the catalytic process due to the excessive H_2_O_2_ that can inhibit the reaction progress. The concentration was restricted to a certain range of Michaelis–Menten curves of two catalysts (Figure 4). It estimated the peroxidase-like activities while keeping H_2_O_2_ or TMB as the unique variable in reactions. It showed that the catalytic activity gradually increased along with the concentration of H_2_O_2_ or TMB, as shown in Figure 4a,b. When the substrate concentration was low, the reaction products showed a rapid growth stage along with the increase in substrate concentration. With the increase in substrate concentration, the change in the absorption peak gradually became slow. In Figure 4c,d, an increase tendency in catalytic property once the concentration of H_2_O_2_ and TMB reached 20 mM and 6 mM, respectively, can be observed. This illustrated the higher quantities of H_2_O_2_ (>20 mM) and resulted in a reaction limit. Due to the no longer increased catalytic activities, in which the competed procedure could be induced while a single catalytic site met the higher concentration of TMB and H_2_O_2_ [50]. Similarly, relative activity rose steadily as TMB concentration rose, eventually reaching a plateau at greater TMB concentrations. In order to investigate the kinetic study of the catalytic process in our work, it was analyzed according to the Michaelis–Menten equation. The apparent steady-state kinetic parameters K_m_ (Michaelis–Menten constant), K_cat_ (turnover number) and V_max_ (the maximal reaction velocity) were set up in further estimation. The lower K_m_, which is equivalent to the substrate concentration in which the rate of conversion is half of V_max_, represents the higher affinity between the catalyst and the substrate. The higher K_cat_, which is obtained by dividing V_max_ by the catalyst concentration, refers the higher rate of enzyme catalysis under the best conditions [53,54]. A series of initial rates (V_0_) of TMB oxidation were calculated from the time-dependent absorption values at 652 nm. The absorption value of the oxidized TMB product was converted to concentration by Equation (1) [39]:(1)$$\left[\mathrm{TMBox}\right]=\frac{A}{\sum_{\left[\mathrm{TMBox}\right]}\times L}$$
where [TMBox] denotes the TMBox concentration, A is the UV-absorbance, L is the solution thickness (1 cm), and $\sum_{\left[\mathrm{TMBox}\right]}$ shows the molar absorption coefficient, having the value of 39,000 M^−1^·cm^−1^ for the TMBox product at 652 nm. By using the above equation we can obtain the value of TMBox.

The plotting of the beginning rates against TMB or H_2_O_2_ concentrations yielded the classic Michaelis–Menten curves. The data were analyzed by the Lineweaver-Burk or double reciprocal plot ($\frac{1}{V_{0}}\;\mathrm{vs}\;\frac{1}{\left[S\right]}$) using Equation (2) [53]. The slop and intercepts of the relevant liner graphs were used to compute the Michaelis–Menten parameters (K_m_) and the maximal reaction velocity (V_max_) values. Further, K_m_ and V_max_ for H_2_O_2_ and TMB were obtained from the reciprocal negative x-intercept and reciprocal y-intercept, respectively. The turnover number (K_cat_) for H_2_O_2_ and TMB were obtained from Equation (3) (Table 1).
(2)$$\frac{1}{V_{0}}=\frac{K_{m}}{V_{\max}\left[S\right]}+\frac{1}{V_{\max}}$$
(3)$$K_{\mathrm{cat}}=V_{\max}/\left[S\right]$$
where $V_{0}$ represents the initial velocity, [S] refers to the concentration of substrate.

Figure 5a,b showed that the peroxidase-like activity of Fe_3_O_4_@PEI@Ag follows a classic Michaelis–Menten enzyme behavior. The double reciprocal of the initial rate (V_0_) versus the substrate concentrations in Figure 5c,d gave an excellent linear relationship. Equations (4) and (5) are the linear regression equations.
(4)$$\frac{1}{V_{0}}=0.918\times 10^{8}\frac{1}{\left[H_{2}O_{2}\right]}+0.077\times 10^{8},{\;R}^{2}=0.993$$(5)$$\frac{1}{V_{0}}=0.260\times 10^{8}\frac{1}{\left[\mathrm{TMB}\right]}+0.086\times 10^{8},{\;R}^{2}=0.988$$

For overall consideration, the comparisons with others enzymatic and HRP catalytic performance as shown in Appendix A [55,56,57,58,59,60]. This assembled Fe_3_O_4_@PEI@Ag structure has high affinity for both target agents due to their huge surface area and multiple active catalytic sites. In Figure 5c,d, the K_m_ values of 1.192 mM and 0.302 mM were obtained for H_2_O_2_ and TMB, respectively. The apparent K_m_ of the Fe_3_O_4_@PEI@Ag structure with H_2_O_2_ as the substrate was about three times lower than HRP, suggesting that the Fe_3_O_4_@PEI@Ag structure have a higher affinity for H_2_O_2_ than HRP. The K_m_ value of Fe_3_O_4_@PEI@Ag structure for TMB was also lower than HRP. This suggested a high affinity and prominent catalytic activity. The advantages of Fe_3_O_4_@PEI@Ag structure in catalytic efficiency were also identified since their K_cat_ values for H_2_O_2_ and TMB were 6.495 × 10^−10^ mol∙s^−1^∙mg^−1^ and 5.815 × 10^−10^ mol∙s^−1^∙mg^−1^. The V_max_ of Fe_3_O_4_@PEI@Ag structure with H_2_O_2_ as substrate (1.299 × 10^−7^ M∙s^−1^) and TMB as the substrate (1.163 × 10^−7^ M∙s^−1^) were higher than HRP (0. 87 × 10^−7^ M∙s^−1^ and 1 × 10^−7^ M∙s^−1^) [61]. By comparing with previous FePt-Au, CuO-Au, GBR, and ZnFe_2_O_4_ [62,63,64] structures, the Fe_3_O_4_@PEI@Ag structure exhibits better enzymatic affinity and peroxidase-like catalytic performance than previously reported metal or metal-based enzyme mimics. The kinetic studies showed that the peroxidase-like activity of the Fe_3_O_4_@PEI@Ag structure is advantageous compared to that of HRP and other previously reported enzyme mimics.

The slope and intercepts of the relevant liner graphs were used to compute the K_m_ and V_max_ values. For overall consideration, the comparisons with other enzymatic and HRP catalytic performances are shown in Appendix A. This assembled Fe_3_O_4_@PEI@Ag structure has high affinity for both target agents due to their huge surface area and multiple active catalytic sites. In Figure 5c,d, it could be seen from Equation (2) that the K_m_ values of 1.192 mM and 0.302 mM were obtained for H_2_O_2_ and TMB, respectively. The K_m_ values were lower for HRP than for the Fe_3_O_4_@PEI@Ag structure treated with H_2_O_2_ and TMB, which suggests its high affinity and prominent catalytic activity. The V_max_ of Fe_3_O_4_@PEI@Ag structure with H_2_O_2_ as substrate (1.299 × 10^−7^ M∙s^−1^) and TMB as the substrate (1.163 × 10^−7^ M∙s^−1^) were higher than HRP (0.87 × 10^−7^ M∙s^−1^ and 1 × 10^−7^ M∙s^−1^) [61]. The specific values could be exactly achieved. By comparing them with the previous FePt-Au, CuO-Au, GBR, and ZnFe_2_O_4_ [62,63,64], the Fe_3_O_4_@PEI@Ag structure exhibit better enzymatic affinity and peroxidase-like catalytic performance than previously reported metal or metal-based enzyme mimics. The kinetic studies showed that the peroxidase-like activity of Fe_3_O_4_@PEI@Ag structure is superior to that of HRP and other previously reported enzyme mimics.

### 2.4. Colorimetric Detection of Glucose and Cytotoxicity Measurement of Fe_3_O_4_@PEI@Ag and Fe_3_O_4_@PEI@Ag@ICG

In clinical diagnosis, the glucose content of urine or blood samples is important in the evaluation of the health or physical condition of a patient. As H_2_O_2_ is the intermediate of the glucose oxidation reaction that is catalyzed by GOx, H_2_O_2_ sensing can provide great assistance to the colorimetric detection of glucose [55]. The mechanism for the detection is as follows:(6)$$D-\mathrm{glucose}+O_{2}\overset{\mathrm{glucose}\;\mathrm{oxidase}}{\to }\mathrm{gluconic}\;\mathrm{acid}+H_{2}O_{2}$$
(7)$$2H_{2}O_{2}+\mathrm{TMB}\overset{{\mathrm{Fe}}_{3}O_{4}@\mathrm{PEI}@\mathrm{Ag}\;}{\to }\mathrm{TMBox}+2H_{2}O+O_{2}$$

The detection of glucose was realized based on the peroxidase-like activity. Fe_3_O_4_@PEI@Ag was evaluated for their potential application as mimics of HRP in glucose detection. Firstly, 100 μL glucose solution was used, with different concentrations ranging from 1 μM to 500 μM in buffer (300 μL, PH 7.4), and these were incubated with glucose oxidase (100 μL, 10 mg∙mL^−1^) at 37 °C for 30 min. Next, the required amount of Fe_3_O_4_@PEI@Ag (100 μL, 1mg∙mL^−1^) and TMB (100 μL, 10 mM) were added in the buffer solutions (300 μL, PH 5.8). A corresponding absorbance (652 nm) would be recorded, and the diagram of colormetric difference exhibited the detection of glucose, eventually (the inset image in Figure 6a). Furthermore, the data was used to draw a linear calibration plot ($y=1.876x+0.008,{\;R}^{2}=0.997$) (Appendix A). Compared to previous references, Fe_3_O_4_@PEI@Ag has a better linear range and low detection limit [65,66,67,68]. Lactose, sucrose, fructose, and maltose were used to investigate the selectivity of this biosensing system, as control samples. Observed from Figure 6b, the glucose (0.5 mM) generated stronger response than the controlled samples (5 mM), reaching a 5-fold higher value.

For verification, cytotoxicity measurement was preliminarily monitored, as shown in Figure 6c,d. The vitality of cells was measured in relation to agent concentration for evaluating cytotoxicity. The effect of Fe_3_O_4_@PEI@Ag and Fe_3_O_4_@PEI@Ag@ICG on Hacat cells was evaluated by MTT assay with different colloidal solutions. There was no reduction of cell viability observed for Hacat cells that were treated with various concentrations (5–200 μg∙mL^−1^) and that underwent incubation for 24 h. Cell viability above 80% means placement in the category of biocompatible materials [69]. Thus, the synthesized Fe_3_O_4_@PEI@Ag structure can be considered as a nontoxic and biocompatible agent.

### 2.5. Photothermal Property of Fe_3_O_4_@PEI@Ag@ICG Structure

The Vis-NIR spectra showed strong and broad absorbance of Fe_3_O_4_@PEI@Ag structure at 700–900 nm (Figure 7a), suggesting its remarkable potential as a photothermal agent. By being coated with photosensitizer, the enhancement of the corresponding photothermal property might be induced. ICG could enhance the original property of assembled structures; the typical absorbance peak of Fe_3_O_4_@PEI@Ag@ICG was at 870 nm (NIR-I, 700–1000 nm). This implied a gradual red-shift by comparing it with pure ICG (780 nm), a change that can be ascribed to the formation of ICG dimers or oligomers, also known as J-aggregates [70]. The intensity was positively correlated with the concentration of Fe_3_O_4_@PEI@Ag@ICG, indicating their mono-dispersion in aqueous conditions (Figure 7b). At the same time, we also explored the near-infrared absorption values of Fe_3_O_4_ NPs and Fe_3_O_4_@PEI@Ag with different concentrations (Appendix A). Their absorption values increased along with concentration increasing, which showed the same phenomenon as before.

An NIR laser of 808 nm (1 W∙cm^−2^) was used to activate the photothermal properties. The Fe_3_O_4_@PEI@Ag@ICG dispersion (200 μL, 200 μg∙mL^−1^) was subject to laser irradiation for 5 min, and the temperature variation was recorded every 20 s in order to evaluate the photothermal phenomenon (Figure 7c). The temperature of the DI water, Fe_3_O_4_ NPs, Fe_3_O_4_@PEI@Ag structure, and Free ICG were increased by 1.6, 4.3, 7.7, and 13.3 °C, respectively, which implied an insignificant photothermal effect. In contrast, a remarkable temperature elevation was observed in Fe_3_O_4_@PEI@Ag@ICG with increments of 21.6 °C under same treatment. This can be attributed to the remarkable photothermal response of the conjugated ICG molecules. Additionally, the temperature elevation of the Fe_3_O_4_@PEI@Ag structure and the Fe_3_O_4_@PEI@Ag@ICG dispersion was concentration-dependent, as shown in Figure 7d and Appendix A. The temperature increased over time in Fe_3_O_4_@PEI@Ag@ICG suspensions, which also exhibited laser output power density-dependent profiles (Appendix A).

The temperature of the Fe_3_O_4_@PEI@Ag@ICG dispersion during laser irradiation was also monitored in real time by infrared imaging (Figure 7e). Such a notable photothermal effect of the Fe_3_O_4_@PEI@Ag@ICG structure may facilitate thermographic imaging in vivo during photothermal therapy (PTT). The stability was evaluated by applying repetitive laser irradiation and cooling (15 min for one cycle) (Figure 7f). Significantly, the Fe_3_O_4_@PEI@Ag@ICG has a faster photothermal response, which can rise to the temperature limit in 3.5 min, and can quickly decrease to RT after removing the Laser. This is more sensitive than that of the widely used PTT agents, such as gold nano shells (5 min and 11 min) [43], copper sulfide (10 min and 10 min) [71], and polydopamine nanoparticles (10 min and 20 min) [72].

## 3. Materials and Methods

### 3.1. Materials

Iron (II) chloride tetrahydrate (FeCl_2_·4H_2_O, 98%) and Sliver acetate (C_2_H_3_O_3_Ag, 99.5%) were purchased from Sigma-Aldrich Co., Ltd. (St. Louis, MI, USA). Iron chloride hexahydrate (FeCl_3_·6H_2_O, 99%), Sodium citrate tribasic dihydrate (C_6_H_5_Na_3_O_7_·2H_2_O, ≥99%), Polyethyleneimine ethylenediamine branched (PEI, average weight 800), PEG−400 (HO(CH_2_CH_2_O)_n_H), average weight 400(MW)), Polyvinylpyrrolidone(PVP, average weight 360,000), Ammonium hydroxide solution (NH_3_·H_2_O), 3,3’,5,5’-Tetramethylbenzidine (TMB, 240.35(MW)), Sucrose (C_12_H_22_O_11_, 342.30(MW)), D-(+)Maltose monohydrate(C_12_H_22_O_11_·H_2_O, 360.31(MW)), D-Fructose(C_6_H_12_O_6_, 180.16(MW)), Glucose Oxidase from Aspergillus niger (GOx, 100 U/mg), Dimethyl sulfoxide(DMSO, 78.13(MW), Phosphate buffer(pH 5.8), phosphate-buffered saline (PBS, pH 7.4), and Indocyanine green (ICG) were supplied by Aladdin Reagent Co., Ltd. (Shanghai, China), D-Glucose(C_6_H_12_O_6_·H_2_O, 198.17) were provided by KESHI (Chengdu, China), α-Lactose(C_12_H_22_0_11_·H_2_O, 360.31(MW)) were obtained from Kermel (Tianjin, China). Ultrapure water utilized throughout the study was produced using a Milli-Q water purification system.

### 3.2. Preparation of Fe_3_O_4_ NPs

Firstly, Fe_3_O_4_ NPs were prepared via a chemical co-precipitation method [42], with slight changes. Briefly, FeCl_2_·4H_2_O (0.19881 g) and FeCl_3_·6H_2_O (0.5406 g) were solved in 100 mL of deionized water with a molar ratio of 1:2 in a triple-neck round-bottom flask. then sonicated for 20 min. The mixture was then heated in a water bath and stirred magnetically. The reaction temperature was then raised to 80 °C, and the solution was allowed to react for 30 min under continuous stirring. In the meantime, when water was heated to 30 °C, Ammonium hydroxide solution (NH_3_·H_2_O) (8 mL) was added. When the temperature reached 40 °C, Sodium citrate tribasic dihydrate (C_6_H_5_Na_3_O_7_·2H_2_O, 2.94 g) was added. The resulting black precipitate, obtained by cooling the reaction mixture to room temperature, was thoroughly rinsed with copious amounts of water several times. The precipitate was separated from the supernatant after each rinsing step using a permanent magnet. Finally, the black-colored precipitate was dried in a vacuum oven at 60 °C overnight.

### 3.3. Preparation of Fe_3_O_4_@PEI@Ag and Fe_3_O_4_@PEI@Ag@ICG Structures

A one-step method was used to prepare the Fe_3_O_4_@PEI@Ag structure, which comprised the formation of silver seeds on the surface of Fe_3_O_4_@PEI NPs and the formation of silver nanoparticles reduced by glucose. Firstly, Fe_3_O_4_ (0.0578 g) and glucose (2.25 g) were dissolved in 40 mL deionized water in a triple-neck round-bottom flask, then sonicated for 20 min. The mixture was heated and stirred in a water bath, and the reaction temperature was then raised to 100 °C. After that, 10 mL PEI aqueous solution (0.1 g∙mL^−1^) was added to the mixture. We also tried to use PVP (10 mL, 0.06 g∙mL^−1^) and PEG (10 mL, 0.2 g∙mL^−1^) as substitutes for PEI. After 30 min, 10 mL of silver acetate was added to the mixture. The reaction temperature was maintained at 100 °C for 2 h with slow magnetic stirring. Afterwards, heating was stopped and stirring continued until the mixture cooled to room temperature. The products were collected with a magnet and washed with Milli-Q water several times. Finally, the black-colored precipitate was dried in a vacuum oven at 37 °C overnight for further use.

A total of 5 mL dispersion of Fe_3_O_4_@PEI@Ag (20 mg∙mL^−1^) was mixed with 20 mL of ICG aqueous solution (100 μg∙mL^−1^) and stirred at room temperature for 24 h under dark conditions. Afterward, reaction products were magnetically collected and wash with DI water several times. The final product was re-dispersed in 5 mL of DI water.

### 3.4. Characterizations

Ultraviolet–visible–near infrared (UV-vis-NIR) diffuse reflectance spectra were employed to investigate the optical properties of different samples on a UV−3600i Plus UV-vis spectrometer. Fourier Transform Infrared (FT-IR) spectroscopy was used to determine the chemical structure of the samples with Thermo Fisher Nicolet iS50 in the range of 400 cm^−1^ to 4000 cm^−1^. Powder X-ray diffraction (PXRD) pattern from 10° to 80° was conducted to characterize the crystal structure of the as-prepared samples using Bruker AXS D8 Advance diffractometer with Cu Kα radiation (v = 2°/min, 40 kV, 40 mA). The morphology and microstructure of different samples were examined by scanning electron microscopy (SEM) on ZEISS Gemini 300 and transmission electron microscopy (TEM) on FEI TF20 Super-X. A vibrating sample magnetometer (VSM) from Quantum Design PPMS−9 was used to measure the magnetic properties of the synthesized nanocomposite. Zeta potential was measured using Nano-Brook Omni (Brookhaven Instruments, Billerica, MA, USA).

### 3.5. Peroxidase-Like Activity Characterization

The peroxidase-like activity of the Fe_3_O_4_@PEI@Ag structure was investigated by using TMB as chromogenic substrates in the presence of H_2_O_2_. The Fe_3_O_4_@PEI@Ag structure reacted with H_2_O_2_ to produce a sufficient amount of ·OH, which was able to oxidize TMB to TMBox blue product, as following Equation (8). We investigated the peroxidase activity by mixing different samples with TMB solution in DMSO and H_2_O_2_. Briefly, samples were prepared in the phosphate buffer (pH 5.8), followed by the addition of H_2_O_2_ and TMB solution. After 10 min, the absorbance change of 652 nm from the oxidation product of TMB was used to monitor all catalytic activities. The pH-dependent peroxidase-like activities were explored by adjusting the pH of phosphate buffer for 10 min before measurements. The relationship between catalytic effect and concentration was verified by varying only the concentration of Fe_3_O_4_@PEI@Ag.

In order to investigate the catalytic activity of the Fe_3_O_4_@PEI@Ag structure, a number of steady-state experiments were conducted by varying concentrations of either H_2_O_2_ or TMB while keeping the remaining parameters constant. The final catalyst concentration of Fe_3_O_4_@PEI@Ag for the kinetics study was 200 μg∙mL^−1^. The aqueous solutions of different samples, H_2_O_2_ solution, and TMB in DMSO were prepared, respectively. For the kinetics study, 100 μL of H_2_O_2_ was fully mixed with 700 μL of phosphate buffer (pH 5.8) and 100 μL of TMB. The additional 100 μL of samples were added to the above mixture to start the catalysis process. After sufficient mixing, UV-vis detection was performed to observe the absorption value of the reaction products at 652 nm. The kinetics were investigated using the below Michaelis–Menten Equation (8).
(8)$$V_{0}=V_{\max}\frac{\left[S\right]}{K_{m}+\left[S\right]}$$
where $V_{0}$ represents the initial velocity, $V_{\max}$ refers to the maximum reaction velocity, and [S] is the concentration of substrate.

### 3.6. Colorimetric Detection of Glucose and Cytotoxicity Measurement of Fe_3_O_4_@PEI@Ag and Fe_3_O_4_@PEI@Ag@ICG Structures

Colorimetric detection of glucose: In clinical diagnosis, the glucose level in urine or blood samples are important for assessing the health of patients. The Fe_3_O_4_@PEI@Ag was evaluated for their potential application as mimics of HRP in glucose detection. H_2_O_2_ is an intermediate product of the glucose oxidation reaction by the action of glucose oxidase. Glucose oxidase reacted with glucose to produce H_2_O_2_, which reacted with Fe_3_O_4_@PEI@Ag in redox reaction and the reaction products further combined with TMB to produce blue TMBox.

(a)A total of 100 μL of GOx and 100 μL glucose with different concentrations in 300 μL Phosphate-buffered Saline (pH 7.4) were incubated at 37 °C for 15 min;(b)A total of 300 μL phosphate buffer (pH 5.8), 100 μL TMB, 100 μL Fe_3_O_4_@PEI@Ag aqueous solutions were added to the above glucose reaction solution (0.5 mL);(c)The mixed solution was incubated at 37 °C for 15 min;(d)UV-vis detection was performed to observe the absorption value of the reaction products at 652 nm. In the control experiments, 10 mM lactose, 10 mM fructose, 10 mM maltose, and 10 mM Sucrose were used instead of glucose, respectively.

Cytotoxicity measurement: The cytotoxic effect of Fe_3_O_4_@PEI@Ag and Fe_3_O_4_@PEI@Ag@ICG structures on human keratinocytes cells (Hacat cells) was determined using the MTT assay. In addition, Hacat cells were obtained from the American Type Culture Collection (Rockville, MD, USA). The cells were stained in accordance with the manufacturer-recommended method and were measured using fluorescence signal analysis equipment. The chemicals used included Dulbecco’s Modified Eagle Medium (DMEM), phosphate-buffered saline, 3-(4,5-dimethyl−2-thiazolyl)−2,5-diphenyl−2H-tetrazolium bromide (MTT, Sigma Aldrich, St. Louis, MI, USA), dimethyl sulfoxide (DMSO, Sigma Aldrich, St. Louis, MI, USA). Measurements were carried out on a multi-detection microplate Synergy HT reader (BioTek, Winooski, VT, USA). We used 96 well plates (Fisher Scientific, Shanghai, China) for cell line cultivation, a centrifugal machine (Eppendorf, Hamburg, Germany), and a glass coverslip (Shitai, Jiangsu, China).

The MTT assay was used to examine the cytotoxic effect and was carried out in six different groups. Briefly, Hacat cells were cultured for 12 h in a 96-well plate (1 × 10^4^ cells per well) and treated with various agents (Fe_3_O_4_@PEI@Ag and Fe_3_O_4_@PEI@Ag@ICG composites) at gradient concentrations (0, 5, 25, 50, 100, and 200 μg∙mL^−1^) at 37 °C and under 5% CO_2_ atmosphere for 2 h. Next, the cells were cultured for another 2 h and subsequently treated with 20 μL MTT solution (500 μg∙mL^−1^ dissolved in PBS). After an additional 4 h incubation at 37 °C and under 5% of CO_2_, the old medium was replaced with 200 μL DMSO and gently agitated for 10 min. The experiment was performed in triplicate for each sample. Finally, the optical absorbance intensity at 490 nm and 630 nm in each well was measured using a microplate reader. The absorbance of each well was measured at 570 nm with an ELISA plate reader.

### 3.7. The Photothermal Effect Measurement under NIR-Light Activations

A NIR laser of 808 nm (LR-MFJ−808/2000 mW, LeiRui, Changchun, China) was used to activate the photothermal effects of the materials. A total of 200 μL of Fe_3_O_4_@PEI@Ag@ICG structure with gradient concentrations were loaded into each transparent Eppendorf (capacity: 1.5 mL) and exposed to NIR irradiation for 5 min. The laser spot was adjusted to cover the whole surface of the samples. The changes in temperature are recorded in real-time by an online thermometer. The photothermal images of different samples were shot every 20 s by an infrared imaging device (FOTRIC 360, Shanghai, China). In order to investigate photothermal stability, one cycle of NIR laser irradiation was applied to the Fe_3_O_4_@PEI@Ag@ICG structure dispersion, and the resultant temperature variations were recorded in real-time during laser irradiation and cooling processes.

## 4. Conclusions

In summary, Fe_3_O_4_@PEI@Ag structure was fabricated via a facile approach in water mediums, the resulting size of which was about 150 nm. The peroxidase-mimicking properties were further explored by TMB, which confirmed that the Fe_3_O_4_@PEI@Ag structure possesses an enzymatic performance with high efficiency due to the synergistic interaction of each composite. Based on the chromogenic reaction of TMB-H_2_O_2_ and GOx, it triggered the generation of H_2_O_2_ from glucose. The catalytic reaction of Fe_3_O_4_@PEI@Ag and colorimetric glucose assay were successfully constructed with excellent sensitivity and selectivity. It could be treated as a replaceable agent to detect glucose in real samples. By integrating it with an ICG photosensitizer, it enhanced the absorbance intensity but, also, Fe_3_O_4_@PEI@Ag@ICG gave red-shift in the NIR-I window. The photothermal property was achieved by irradiation at 808 nm, which showed the synergistic effect and excellent performance at high efficiency in temperature-controlled conditions. The Fe_3_O_4_@PEI@Ag@ICG structure has a faster photothermal response, which can rise to the temperature limit in 3.5 min and quickly decrease to RT within 10 min once the laser source is removed. It should be considered as a potential path for making complex composites, and the subsequent development could offer the possibility of applying this structure to photothermal therapies.

## Data Availability

The data presented in this study are available on request from the corresponding author.

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
