# Peer review of "New Insight into Assembled Fe3O4@PEI@Ag Structure as Acceptable Agent with Enzymatic and Photothermal Properties"

_ijms, 2022, doi:10.3390/ijms231810743_

Round 1

Reviewer 1 Report

The major problems with this manuscript are very poor English and style, hiding in many instances the intended meaning. The authors clearly need help from a professional editing service. However, the scientific content is quite interesting, and the manuscript can be reconsidered after editing/rewriting.

The authors refer to their synthetic materials as "enzymes". A more appropriate term could be "enzyme mimics" as all known enzymes are naturally occurring polypeptides.

Kinetic data presentation and analysis are nonprofessional. It remains unclear how the kinetic parameters were derived, as the line in Fig 5b intersects the ordinate in the negative range.

Vmax is not a proper characteristic of the catalytic power as it depends on the amount of the catalyst. A specific activity (Vmax divided by catalyst amount) would be more appropriate, like in enzymology.

Some figure legends do not provide sufficient information to understand the figure.

Reviewer 2 Report

Authors report an inorganis framewokr active as an enzyme for use in medical applications.

The article is well written, except for some typos, e..g feild.

I suggest adding a scalebar to the microscopy image and enrich the figure captions with more experimental details, to make the figure more self-explanatory.

Statistical analysis can be improved.

A comparative table between the biochem features of a traditional enzyme and the iron-Ag framework would help to understand the value.

Round 2

Reviewer 1 Report

The revised manuscript addresses two of five previous remarks. The major problem that remained unaddressed is very poor English and style, hiding in many instances the intended meaning. There are so many style, word choice, and grammar problems that the manuscript should be completely rewritten. To characterize the degree of the required revision, I would say that nearly every third word should be changed/modified. The authors clearly need help from a professional editing service.

As I previously mentioned, Vmax is not a proper characteristic of the catalytic power as it depends on the amount of the catalyst. To compare different catalysts, one should make sure that same amounts of the catalysts are used to measure reaction rate. This is accomplished by comparing specific activities (Vmax divided by catalyst concentration), whose unit will be mol·s-1·mg-1, like in enzymology.

The authors should check that sufficient experimental details are provided for each experiment to allow its exact reproduction by the reader.

Round 3

Reviewer 1 Report

The language and style changes made in manuscript do not improve its quality cardinally. The manuscript is still clearly in need of a professional editing.

Author Response

Reviewer’s Comments and Suggestions for Authors:

The language and style changes made in manuscript do not improve its quality cardinally. The manuscript is still clearly in need of a professional editing.

--We really appreciated for reviewer’s comments in the language modification. Anyway, it is the one essential promotion in corrections. We have modified this, please check it (track changes parts).
